# Ischemic Preconditioning Promotes Post-Exercise Hypotension in a Session of Resistance Exercise in Normotensive Trained Individuals

**DOI:** 10.3390/ijerph17010078

**Published:** 2019-12-20

**Authors:** Patricia Panza, Jefferson Novaes, Luiz Guilherme Telles, Yuri Campos, Gleisson Araújo, Nacipe Neto, Leandro Raider, Giovanni Novaes, Luis Leitão, Jeferson Vianna

**Affiliations:** 1Faculty of Physical Education and Sports, Federal University of Juiz de Fora, Juiz de Fora-MG 36036-900, Brazil; paty_panza@yahoo.com.br (P.P.); jeffnovaes@gmail.com (J.N.); reiclauy@gmail.com (Y.C.);; 2Faculty of Physical Education and Sports, Federal University of Rio de Janeiro, Rio de Janeiro-RJ 21941-901, Brazil; guilhermetellesfoa@hotmail.com (L.G.T.); profgleissonon@hotmail.com (G.A.); 3Physical Education Department, Estácio de Sá University, Rio de Janeiro-RJ 22080-000, Brazil; 4Faculty of Medical and Health Sciences of Juiz de Fora, Suprema, Juiz de Fora-MG 36033-003, Brazil; dr.nacipejacob@gmail.com; 5Division of Endocrinology, IPEMED Medical School, São Paulo-SP 04143-020, Brazil; 6Faculty of Physical Education, Center of Higher Education of Valença, University Center of Valença, Valença 27600-000, Brazil; leandroraider@raideracademia.com.br; 7Brazilian Music Consertatory, Brazilian University Center of Education, Rio de Janeiro-RJ 20520-140, Brazil; giovanninovaes@gmail.com; 8Superior School of Education, Polytechnic Institute of Setubal, Setubal 2910-761, Portugal

**Keywords:** ischemic preconditioning, resistance exercise, blood pressure, post-exercise hypotension

## Abstract

Ischemic preconditioning (IPC) is a method that has been used prior to resistance exercise to improve performance. However, little is known about its effect before a resistance exercise training session on hemodynamic responses. Thus, the aim of the study was to verify the acute effect of IPC before a session of resistance exercises on the systolic blood pressure (SBP), diastolic blood pressure (DBP), and mean blood pressure (MBP) of trained normotensive trained individuals. Sixteen men (25.3 ± 1.7 years; 78.4 ± 6.2 kg; 176.9 ± 5.4 cm, 25.1 ± 1.5 m^2^.kg^−1^) trained in resistance exercise (RE) (5.0 ± 1.7 years) were evaluated in five sessions on non-consecutive days. The first two sessions’ subjects performed one repetition maximum (RM) test and retest, and for the next three sessions, they performed the experimental protocols: (a) IPC + RE; (b) SHAM + RE; (c) RE. The RE protocol consisted of six multi-joint exercises, three sets at 80% of 1RM until concentric failure. Blood pressure was monitored pre-session, immediately after and every 10 min for 60 min after RE. IPC consisted of 4 × 5 min of vascular occlusion/reperfusion at 220 mmHg. SHAM (fake protocol) consisted of 20 mmHg of vascular occlusion/reperfusion. The IPC + RE protocol showed significant reductions on SBP, DBP, and MBP compared with SHAM + RE (*p* < 0.05) and with RE (*p* < 0.05). The IPC + RE protocol presented a greater magnitude and duration of post-exercise hypotension (PEH) from 20 to 60 min after exercise in SBP (−11 to 14 mmHg), DBP (−5 to 14 mmHg), and MBP (−7 to 13 mmHg). Therefore, we can conclude that the application of IPC before an RE session potentiated the PEH in normotensive individuals trained in resistance exercise.

## 1. Introduction

Resistance exercise has been recommended as a nonpharmacological intervention to prevent and treat cardiovascular diseases [1,2]. Several studies [3,4,5,6,7,8,9] have shown benefits on the regulation of blood pressure (BP) in normotensive trained individuals and suggested resistance exercise as an important strategy to control blood pressure to promote cardiovascular health in normotensive or hypertensive participants [10,11] and to promote acute reductions in post-exercise blood pressure [1,2,3,4,5,6,7,8,9,10,11,12]. This phenomenon called by Kenney and Seals [13] as post-exercise hypotension (PEH) can last up to 24 h and is characterized by a reduction in systolic blood pressure (SBP) or diastolic blood pressure (DBP) after a single exercise session below rest baseline or pre-exercise level [12,13]. The literature [12,14] shows that PEH has great clinical relevance for the treatment of hypertensive patients, as well as for prevention in normotensive individuals. A reduction in resting blood pressure of 3 mmHg is associated with 8% less chance of stroke mortality and 5% less mortality from coronary heart disease [12]. Although resistance exercise has traditionally been used to promote PEH [3,4,5,6,7,8,9], its effects have not yet been fully elucidated [15], and further investigations are needed when the resistance exercise is associated with vascular occlusion prior to exercise.

Ischemic preconditioning (IPC) is a technique of vascular occlusion alternated with moments of reperfusion, applied through a pneumatic tourniquet before performing an exercise in a non-invasive manner [16]. The application of IPC can increase blood flow in skeletal muscles [17], liver [18], heart [19], and kidneys [20]. This intervention was initially developed to reduce the myocardial damage caused by sustained ischemia [21]; however, IPC affects exercise performance [16,22,23] by improving muscle oxygenation and blood flow to active tissues and organs [24].

Based on the hypothesis that IPC may increase heart blood flow and post-reperfusion muscle performance, skeletal muscle research has been initiated by associating IPC with physical exercise [16,22,23,24]. A few studies have verified the effect of IPC on blood pressure response after exercise, and some studies have verified the effects of IPC applied before cycling tests [22,23] and in running [25], but did not present significant acute reductions in post-exercise blood pressure. However, the analysis of blood pressure occurred only between the anterior and immediately posterior moments, and the hypotensive effect was not evaluated. Furthermore, other studies have shown important clinical effects of IPC on blood pressure, which seems to be a new and efficient therapeutic strategy in the treatment of arterial hypertension [26]. Madias [27] evaluated the effect of IPC on the blood pressure of a normotensive individuals for 60 min and demonstrated that IPC reduced blood pressure after the application of three sets of 5 min cycles of vascular occlusion (VO), alternating with three cycles of reperfusion in the resting arms. Jones et al. [28] applied seven consecutive days of IPC in young normotensive patients and observed a reduction in mean blood pressure (MBP), and Battipaglia et al. [29] observed a reduction in SBP and rate pressure product [HR (beats·min-1) × SBP (mmHg)] immediately after performing ergometric tests in patients with coronary artery disease. In addition, recently, Tong et al. [26] reported chronic blood pressure reduction in hypertensive individuals with IPC application for 30 days. However, no study verified the effects of IPC applied before the resistance exercise session on the PEH.

The mechanisms of PEH in resistance exercise are mediated by the reduction of systemic vascular resistance, decreased sympathetic activity, increased baroreflex activity, and vasodilation [13]. On the other hand, the effects of IPC are mediated by the phosphorylation of the enzyme nitric oxide synthase and consequently the release of nitric oxide in the circulation, causing vasodilation [30], increased blood flow and arterial diameter [25], and increased parasympathetic activity in humans [31]. Therefore, we hypothesized that IPC when applied before the resistance exercise session can potentiate the PEH. The aim of the study was to verify the acute effect of IPC before a session of resistance exercise on the SBP, DBP, and MBP of trained normotensive individuals trained in RE. In addition, this research may help strength and conditioning coaches and practitioners during the prescription and selection of training methods with the goal to improve the hemodynamic responses in normotensive individuals.

## 2. Materials and Methods

### 2.1. Experimental Design

The present study was conducted during 5 sessions on non-consecutive days (3 days apart), always at the same time of the day to avoid circadian influence. During the first visit to the laboratory, the participants signed an informed consent form and answered the Physical Activity Readiness Questionnaire (PAR-Q), and immediately after, anthropometry assessments and one repetition maximum test (1RM) were performed. In the second session, a 1RM retest was performed for load reproducibility, and from the third to the fifth visit to the laboratory, the participants were randomly assigned to the following experimental protocols: (a) resistance exercise at 80% of 1RM session protocol (RE); (b) IPC + RE at 80% of 1RM (IPC); (c) SHAM + RE at 80% of 1RM (SHAM). SBP, DBP, and MBP were monitored before, immediately after, and 10, 20, 30, 40, 50, and 60 min after exercise.

### 2.2. Subjects

Sixteen normotensive men (25.3 ± 1.7 years; 78.4 ± 6.2 kg; 176.9 ± 5.4 cm; 25.1 ± 1.5 m^2^.kg^−1^) with at least three years of experience in RE (5.0 ± 1.6 years) were included in the study. To recruit the sample, the procedures suggested by Beck [32] were adopted. The sample size was calculated using G*Power 3 software. Based on a previous analysis, an n of 16 subjects was calculated after using a power of 0.80, α = 0.05, a correlation coefficient of 0.5, the nonsphericity correction of 1, and an effect size of 0.32. It was verified that the sample size was sufficient to provide 83.8% of the statistical power. For calculation of the sample, the procedures suggested by Beck [32] were adopted. Subjects were excluded from the study if they responded positively to any of the items in the Physical Activity Readiness Questionnaire [33], missed one of the sessions of the collection procedures in the laboratory, presented some type of musculoskeletal injury in the upper or lower limbs, had smoking habits, obesity, hypertension, supplements’ consumption, or were on medications. After explaining the risks and benefits of the research, the subjects signed an informed consent form elaborated according to the Declaration of Helsinki. The study complied with Resolution 466/12 of the National Health Council and was approved by the local Research Ethics Committee of the University Center of Volta Redonda under Protocol Number 2,699,294.

### 2.3. Resistance Exercise Session

The resistance exercise session was performed at 80% of 1RM. Warm-up was performed first with two sets of 15 repetitions at 50% of 1RM, with an interval of 1 min between sets. The resistance exercise session consisted of six exercises alternating upper and lower body, namely: bench press (BPr), 45° leg press (LP), lat pull down (LPD), hack machine (HM), shoulder press (SP), and Smith squat (SS) (Buick © Fitness Equipment, Rio de Janeiro, Brazil). Each exercise was performed with a volume of three sets with 80% of 1RM, until concentric failure, with an interval of 1 min and 30 s between sets and two minutes between exercises. The experimental design of the study can be observed in Figure 1.

### 2.4. IPC and SHAM Experimental Protocols

The IPC protocol consisted of 4 occlusion cycles of 5 min at 220 mmHg of pressure, using a 57 cm × 9 cm pneumatic tourniquet around the upper arm sub-axillary region (Riester Komprimeter^®^, Jungingen, Germany) alternating 5 min of reperfusion at 0 mmHg [15,33], resulting in a total intervention of 40 mins. The pressure and the cuff width used were in line with previous studies [16,34], and the blood flow obstruction was verified by digital palpation.

The SHAM protocol session consisted of 4 sets of 5 min occlusion cycles at 20 mmHg of pressure, as proposed in previous studies [16,34], alternating with 5 min of reperfusion at 0 mmHg for a total of 40 min of intervention. Occlusion and reperfusion cycles were applied alternately between the right and left arms, with the subjects sitting during the entire protocol [34].

### 2.5. Procedures

#### 2.5.1. Anthropometric Evaluation

Height and body mass were measured with a precision of 0.5 cm and 0.1 kg, respectively, through a Filizola^®^ stadiometer and scale; the body mass index (BMI) and all measurements were taken following the recommendations of the American College of Sports Medicine [10].

#### 2.5.2. RM Test

Training load prescription was based in the 1RM evaluation [10] performed in the 1^st^ and 2^nd^ visits to the laboratory. The BPr, LP, LPD, HM, SP, and SS exercises were performed bilaterally, with a 10 min pattern for recovery between exercises. For warm-up, each individual performed two sets of 5–10 repetitions at 40–60% (1 min interval between sets), of the individual’s maximum strength perception. After 1 min of rest, a third set was completed between 3 and 5 repetitions at 60–80% of maximum perceived force. After another resting period (1 min), the strength evaluation started with 5 attempts, adjusting the load before each new attempt. The recovery duration between attempts was standardized to 3–5 min. The test was interrupted when the individual could not execute the movement correctly, the maximum load being considered the one used in the last correct repetition. The following strategies were adopted to reduce the margin of error in the data collection procedures: (a) standardized pre-test instructions, so that each subject would be aware of the entire routine involved in data collection; (b) the subject was instructed on the appropriate exercise execution; (c) all participants received standardized verbal encouragement during the tests; and (d) all tests were performed at the same time of the day for each session. The highest load achieved between the two days was considered the 1RM.

#### 2.5.3. Blood Pressure Monitoring

An automatic blood pressure monitor Microlife^®^ BP3BTO-A was used, previously validated according to the British Association of Cardiology criteria for resting measurements [35]. The cuff was placed on the left arm and was completely wrapped, covering at least two thirds of the upper arm. This equipment was used for all pre- and post-session blood pressure measurements. The measurements were made with subjects seated during the first 10 min of rest, immediately after exercise and every 10 min for a period of 1 h after exercise. All measurements were performed according to the American Heart Association guidelines [36] and were performed at the same time of day to avoid the interference of the circadian rhythm in blood pressure. SBP and DBP were measured, and MBP was calculated using the [systolic blood pressure + (2 X diastolic blood pressure)]/3 equation. These measurements were performed by a technician who was blind to the experimental protocol previously applied.

### 2.6. Statistical Analysis

The results were presented with the mean ± standard deviation. To test the normality and homogeneity of the data, the Shapiro–Wilk test and the Levene test were performed. A two-way repeated measures ANOVA was performed to determine differences between experimental protocols and moments of assessment. The Bonferroni post hoc test was also performed. The effect size (ES) was used to determine the magnitude of changes between experimental protocols. The magnitude of the ES was interpreted using the scale proposed by Rhea [37]. Statistical analyses were performed using the SPSS 21^®^ statistical software package (SPSS Inc., USA), adopting a critical significance level of *p* < 0.05.

## 3. Results

All variables presented a normal distribution (*p* < 0.05). The intraclass correlation coefficient (ICC) was used to evaluate the 1RM reproducibility (ICC, BPr = 0. 98, LP = 0.99, LPD = 0.98, HM = 0.97, SP = 0.98, SS = 0.97). The effect size, *p*-values, and percentage changes (Δ%) for each condition and time point are presented in Table 1.

Significant differences were found in the comparison across the different experimental conditions for SBP, DBP, and MBP. A significant protocol × time interaction showed increases compared with the baseline for DBP (F_(14, 315)_ = 2.707; *p* = 0.0009) and decreases compared with the baseline for SBP and MBP (SBP: F_(14, 315)_ = 2.198; *p* = 0.0078; MBP: F_(14, 315)_ = 3.176; *p* = 0.0001) in IPC + RE, SHAM + RE, and RE (Figure 2).

## 4. Discussion

The present study investigated the acute effect of IPC applied before a resistance exercise session on SBP, DBP, and MBP of trained normotensive youth. This is the first study that has investigated the post-exercise response of blood pressure combining the application of IPC before a session of high-intensity resistance exercise (80% of 1RM) for upper and lower limbs (alternated by segment). The main findings were that IPC + RE presented significant reductions on SBP, DBP, and MBP, when compared with the SHAM + RE and RE protocols, confirming our hypothesis that the application of IPC before resistance exercise can potentiate PEH. The IPC + RE protocol presented great reductions of –11 to 14 mmHg on SBP, –5 to 14mmHg on DBP, and 7 to 13 mmHg on MBP. The SHAM + RE protocols presented reductions post exercise of 3 to 5 mmHg on MBP and the RE protocol of –5 to 7 mmHg on SBP, ~8 mmHg on DBP, and –5 to 7 mmHg on MBP.

Our results were in accordance with the literature [27,28,29,38] that the addition of IPC before a resistance exercise session potentiated the PEH. Few studies confirmed the PEH after single application of the vascular occlusion technique [27,28,38], and only one study verified chronic blood pressure reduction [26]. In the study of Madias [27], the IPC technique was applied for three sessions, on non-consecutive days, with 3 cycles of 5 min of ischemia alternated by 3 cycles of 5 min of reperfusion. The results indicated, 30 min after the technique, a significant mean reduction of 6 mmHg in SBP, 3 mmHg in DBP, and 3 mmHg in pulse pressure (PP). Luca et al. [38] showed that the application of the IPC technique for one day promoted and for seven days sustained protection against ischemia-reperfusion injury caused by endothelial dysfunction. Jones et al. [28] applied the IPC technique in four cycles of 5 min of ischemia/reperfusion with 220 mmHg in the arms for seven consecutive days in normotensive young individuals. The authors observed a 3 mmHg reduction in MBP immediately after seven days of intervention and a 5 mmHg reduction after 14 days of intervention. In addition, there was an increase in local and remote blood flow after 14 days of intervention. Recently, Tong et al. [26] observed a chronic reduction of blood pressure at an average of 8 mmHg for SBP and 6 mmHg for DBP with the application of the technique using three cycles of 5 min of ischemia/reperfusion for 30 consecutive days in hypertensive patients.

The IPC is a new and promising therapeutic intervention promoting an acute [27] and chronic effect [26] on blood pressure reduction, similar to the acute and chronic blood pressure reduction caused by physical exercise in general [10]. Some physiological mechanisms reported in previous studies [26,39], such as the release of humoral factors in the local and systemic circulation, which leads to the activation of neural pathways [26] by the release of adenosine, bradykinin, calcitonin, and opioids [39], may have influenced the results of our study. Recently, research has focused on the release of protective humoral factors, such factor 1α derived from stromal cells (SDF-1α), nitrite, exosomes, and interleukin [26]. SDF-1α seems to be an important chemokine capable of modulating endothelial progenitor cells that act on endothelial repair after an ischemia/reperfusion injury [40]. Tong et al. [26] observed a circulating SDF-1α increase after 30 days of daily application of IPC with three cycles of 5 min with 200 mmHg in the arms accompanied by chronic reduction of SBP and DBP and improvement of endothelial function in hypertensive individuals.

Several studies have demonstrated PEH in a traditional resistance exercise session [3,4,5,6,7,8,9] in trained individuals. The SBP response in the present study presented significant PEH in the RE and in the IPC + RE protocols. Similar results were reported by Duncan et al. [5], who compared different intensities (40 and 80% 1RM) in a resistance exercise session and demonstrated significant PEH on SBP only for the high intensity protocol. Rezk et al. [9], comparing RE at 40 and 80% 1RM, also found significant reductions in SBP in the 80% 1RM protocol. Polito et al. [8] compared two resistance exercise sessions at different intensities (6RM and 50% of 6RM) and observed significant reductions in SBP for both protocols. However, only the low intensity protocol showed a significant reduction in DBP. It seems that high intensity, regardless of ischemic preconditioning, level of training, and gender, promotes hypotensive effect in SBP after resistance exercise in most studies [3,4,5,6,7,8,9]. The decreases in preload and the reduction of systolic volume after high intensity resistance exercise [9] is one possible mechanism that may have influenced the results of the PEH in SBP in our study. This can occur because the plasmatic volume is reduced by transporting the plasma liquid from the blood to the interstitial space [41], and as the magnitude of this plasma reduction is greater after more intense exercises [41], this mechanism may have influenced our results regarding the greater reduction in SBP after the IPC + RE session. However, the decrease in plasma volume and, consequently, in venous return may also be involved in the increase of peripheral vascular resistance, as observed in the study by Rezk et al. [9]. This decrease may disable cardiopulmonary receptors, increasing peripheral sympathetic activity and peripheral vascular resistance. However, IPC may have represented an additional stimulus in the neural and humoral pathways [39] that may have reduced the peripheral vascular resistance by vasodilation promoted by the technique itself [30]. In sum, these mechanisms may have been responsible for the greater magnitude and duration of the PEH in the IPC + RE protocol over SBP when compared to the SHAM + RE and RE protocols.

Regarding the DBP response, our findings demonstrated a significant PEH in the IPC + RE and RE protocols. However, in the SHAM + RE protocol, we observed an important clinical reduction (~5 mmHg) of DBP over 60 min. Furthermore, the SHAM + RE protocol that used 20 mmHg of vascular occlusion pressure presented a significant PEH post-20 min and post-40 min on MBP. Some studies on high intensity resistance exercise demonstrated a reduction in PEH for DBP [3,7], mainly due to the increase in peripheral vascular resistance caused by the training adaptations [9]. Keese et al. [7] performed a resistance exercise session with 80% of 1RM and reported a reduction on DBP only for 20 min post-exercise. Bentes et al. [3] compared two different intensities (60 and 80% 1RM) and found no significant differences. The greater PEH verified in our study with the application of the vascular occlusion can be explained by the increased secretion of nitric oxide [30], arterial diameter, and blood flow [25,38]. These may have enhanced post-exercise vasodilatation by reducing peripheral vascular resistance [9], which may have reduced DBP, even after a high intensity session.

Therefore, we can conclude that the application of IPC before a resistance exercise session may enhance the PEH in SBP, DBP, and MBP in normotensive individuals trained in resistance exercise. Additionally, because this study used normotensive young subjects, the results may not be applied to other populations, such as hypertensive subjects. It is recommended that more studies be conducted to elucidate the effects of IPC before a resistance exercise session on physiological and autonomic variables, in order to verify cardiac output, peripheral vascular resistance, and heart rate variability, mainly including subjects of different levels of conditioning, age, and clinical conditions.

## 5. Conclusions

Ischemic pre-conditioning represents an additional stimulus to the high intensity resistance exercise session, generating a greater hypotensive effect. We suggested that ischemic preconditioning could be used by health and exercise professionals when prescribing high intensity resistance exercise with six multi-joint exercises, three sets at 80% of 1RM performed until concentric failure. However, the results of this study are likely to apply only to normotensive trained male adults, and further research testing other populations, including hypertensive individuals, is needed.

## Figures and Tables

**Figure 1 ijerph-17-00078-f001:**
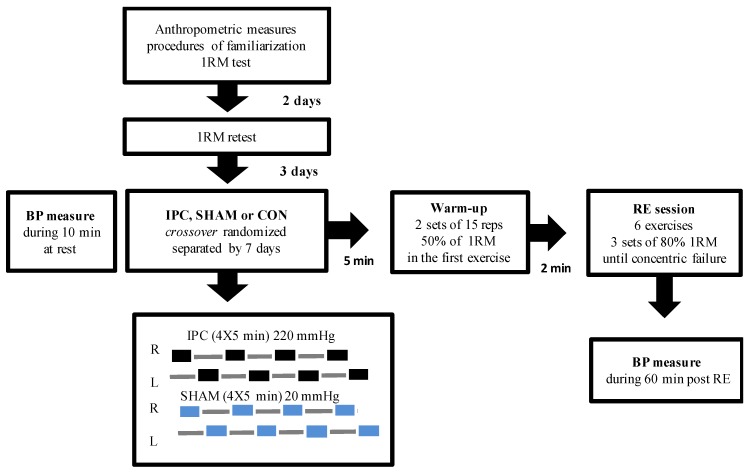
Experimental design of the study; 1RM: one repetition maximum; IPC: ischemic preconditioning; SHAM: false protocol; RE: resistance exercise; CON: control protocol; BP: blood pressure.

**Figure 2 ijerph-17-00078-f002:**
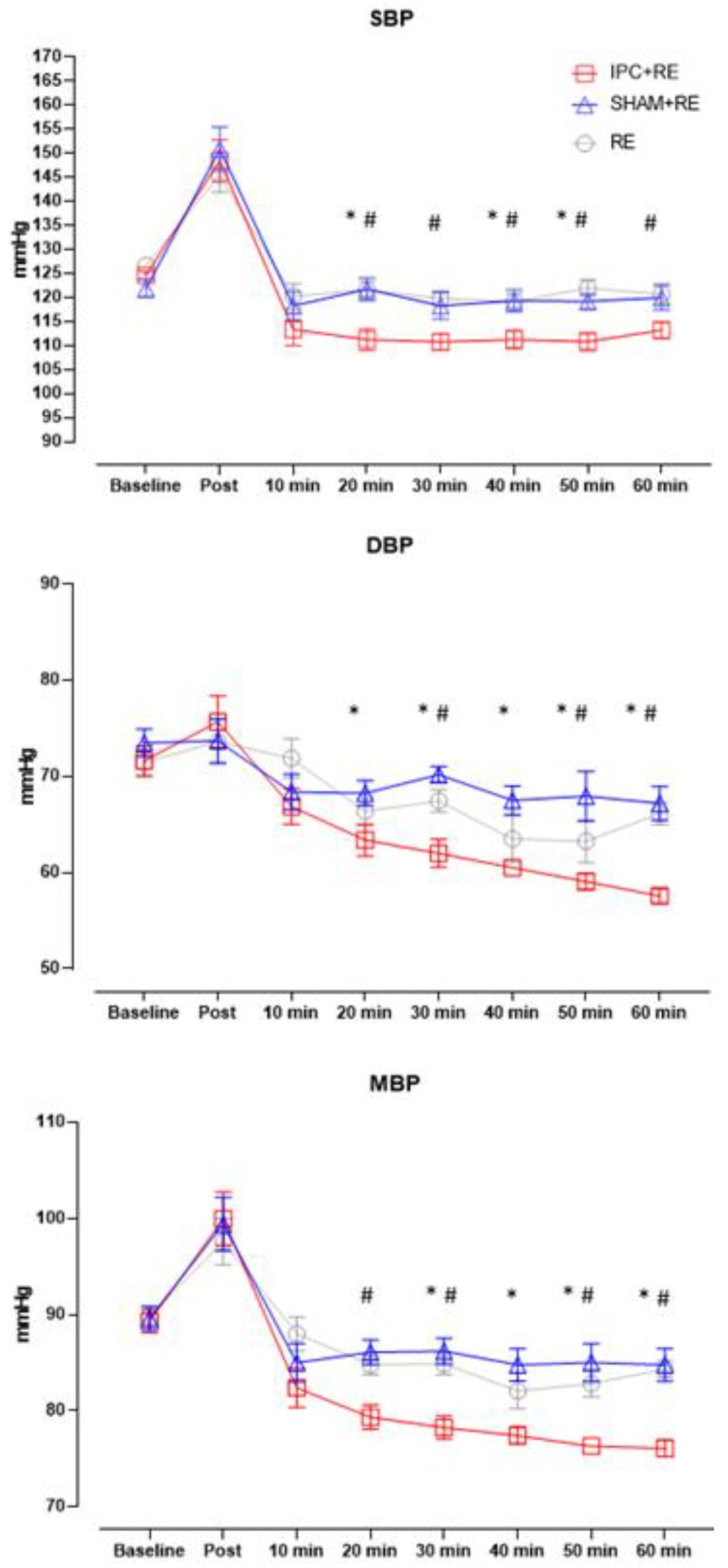
Blood pressure response after all experimental protocols. Resistance exercise protocol (RE); ischemic preconditioning protocol (IPC + RE); false protocol (SHAM + RE); ^#^ significant difference between IPC vs. RE (*p* < 0.05); * significant difference between IPC vs. SHAM (*p* < 0.05).

**Table 1 ijerph-17-00078-t001:** Ischemic preconditioning protocol (IPC + RE); fake protocol (SHAM + RE); resistance exercise protocol (RE); ES = effect size; Δ% = difference between post and baseline moments in percentage; systolic blood pressure (SBP); diastolic blood pressure (DBP); mean blood pressure (MBP).

	IPC + RE			SHAM + RE			RE		
	ES	Δ%	*p*	ES	Δ%	*p*	ES	Δ%	*p*
SBP									
Post-10	−3.15	−9.0	0.01	−0.49	−3.0	0.80	−1.47	−5.1	0.55
Post-20	−3.74	−10.7	0.02	−0.02	−0.1	1.00	1.15	−4.0	0.04
Post-30	−3.88	−11.1	0.01	−0.49	−3.0	0.80	1.57	−5.4	0.01
Post-40	−3.74	−10.7	0.02	−0.35	−2.1	0.96	1.71	−5.9	0.01
Post-50	−3.86	−11.1	0.02	−0.36	−2.2	0.95	−1.08	−3.7	0.20
Post-60 DBP	−3.20	−9.2	0.05	−0.26	−1.6	1.00	−1.37	−4.7	0.29
Post-10	−0.77	−6.6	0.01	−0.89	−7.0	0.42	0.07	0.6	1.00
Post-20	−1.40	−11.5	0.01	−0.91	−7.1	0.39	0.86	−7.1	0.44
Post-30	−1.57	−13.4	0.02	−0.58	−1.7	0.87	0.68	−5.6	0.73
Post-40	−1.81	−15.5	0.01	−1.04	−8.2	0.22	−0.82	−11.1	0.01
Post-50	−2.05	−17.5	0.01	−0.97	−7.6	0.31	−0.94	−11.5	0.01
Post-60 MBP	−2.23	−19.6	0.01	−1.10	−8.6	0.17	0.90	−7.4	0.17
Post-10	−1.48	−7.7	0.53	0.63	−5.2	0.11	−0.37	−2.1	1.00
Post-20	−2.14	−11.2	0.01	0.74	−4.0	0.01	−1.24	−5.6	0.02
Post-30	−2.37	−12.9	0.01	0.71	−3.8	0.15	−1.21	−5.5	0.01
Post-40	−2.55	−13.3	0.01	1.01	−5.4	0.01	−1.90	−8.7	0.02
Post-50	−2.79	−14.5	0.01	0.96	−5.1	0.35	−1.72	−7.8	0.01
Post-60	−2.83	−14.8	0.01	1.01	−5.4	0.09	−1.35	−6.2	0.01

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
