# Peer review of "Ischemic Preconditioning Promotes Post-Exercise Hypotension in a Session of Resistance Exercise in Normotensive Trained Individuals"

_ijerph, 2019, doi:10.3390/ijerph17010078_

Round 1

Reviewer 1 Report

The manuscript focuses on a relevant research theme, although with not many studies carried out particularity regarding the effects of IPC before the RE on PEH, which makes it an original and a pertinent paper. The study design is solid and fully suited to the defined objectives as well as the methodology used. Congratulations to the authors. No further suggestions.

Author Response

We are grateful for your consideration of this manuscript, and we also very much appreciate your suggestions, which have been very helpful in improving the manuscript.

Reviewer 2 Report

In my opinion, the changes made to the paper are not sufficient to make it suitable for publication.

The English language is a serious concern compromising the understanding of this manuscript. Some of the statistical symbols shown in the tables were changed and unfortunately made the data interpretation even more difficult: Is it really possible to significantly change mean blood pressure without changing systolic and diastolic blood pressure (SHAM + RE group)? Since mean blood pressure was calculated by the equation: [systolic blood pressure + (2 X diastolic blood pressure)] / 3, it seems mathematically weird.  The statistical symbols presented in the added figure are not consistent with the symbols shown in the tables. 

Author Response

We are grateful for your consideration of this manuscript, and we also very much appreciate your suggestions, which have been very helpful in improving the manuscript. The comments are addressed below.

REVIEWER: In my opinion, the changes made to the paper are not sufficient to make it suitable for publication. The English language is a serious concern compromising the understanding of this manuscript. Some of the statistical symbols shown in the tables were changed and unfortunately made the data interpretation even more difficult: Is it really possible to significantly change mean blood pressure without changing systolic and diastolic blood pressure (SHAM + RE group)? Since mean blood pressure was calculated by the equation: [systolic blood pressure + (2 X diastolic blood pressure)] / 3, it seems mathematically weird.  The statistical symbols presented in the added figure are not consistent with the symbols shown in the tables.

RESPONSE- We rewrote the article and improved the English language. In the results, we add a table with all data and improved the discussion. We have redone the calculation of MBP, which was really miscalculated.

Reviewer 3 Report

Thank you for the opportunity to review this manuscript.

This study investigated the effect of ischemic preconditioning on post-exercise hypotension in healthy young men.  Blood pressure typically decreases after a bout of resistance training and this study examined whether ischemic preconditioning would increase this response.

General comments:

Throughout this manuscript, there are opportunities to make the message clearer by improving the word choice, avoiding repetition, and correcting grammatical errors. In addition, I would suggest that overuse of abbreviations interferes with readability, and therefore, understanding. Was there any effort to blind the investigators to the condition during post-exercise blood pressure measurement? Given the hypothesis presented in the introduction and the confirmation of the hypothesis stated in the discussion, it would seem prudent to indicate some management for bias in the data. The results section needs work. First, the same data are presented in tables and figures.  This is redundant.  The text includes lengthy lists of individual changes, effect sizes, and significance.  Consider presenting these findings in a table with the data.  If the figures are to be used, please consider how you indicate significance.  For example, you may include one marker to indicate a significant difference to other conditions and one marker to indicate a difference from baseline (if this is important).  Four markers of significance for individual data points suggests inappropriate statistical analysis (too many individual comparisons). The discussion lists individual findings from other papers. It would be helpful if these findings were grouped or summarised and, importantly, put into context relative to the findings in this study.  This is done well in lines 306-308, but not done so well elsewhere in the manuscript. There are a few instances of ideas that should be suggestions rather than statements. For example, stating that your results “can be explained by xyz”, suggests you have proof of the mechanism.  I suggest alternative phrasing, such as “xyz may contribute to these findings” or “one mechanism that may have influenced this result is”…

Specific comments:

Title: consistent use of capital letters

Line 25: spaces after commas and format units

Line 25-26: five visits on non-consecutive days – current wording is confusing

Lines 40-44: be concise and specific, plus needs grammatical editing

Line 46: phrasing

Line 49: prevention of normotensive individuals?

Line 54: please avoid starting sentences with abbreviations

Line 54-55: phrasing

Line 61-62: phrasing

Line 64: do you mean they did not present their data or their data showed the intervention had no effect?

Line 68: normotensive individuals (plural)

Line 72: double product?

Line 85: inconsistent use of abbreviations – I’d encourage you not to abbreviate resistance exercise throughout the manuscript

Line 89: experimental design

Line 95-96: volunteers were randomly assigned

Lines 100-101: Sixteen normotensive men (tidy up units) with at least three years experience…

Lines 103-111: this section needs to be reviewed

Figure 1: were the warm up sets carried out for all exercises?

Line 168: check your equation

Line 173-174: ICC results should be presented in the results section, rather than in the description of statistical methods

Line 261-264: phrasing could be improved for clarity

Line 265-266: this is repetitive

Line 283-285: rephrase for clarity

Line 286-292: this explanation needs to be tidied up

Line 306-308: This is well done.  This is a nice example of putting other research findings into context to help explain your own findings.  Please do more of this throughout your manuscript.

Line 341: interesting does not fit here

I wish you the best for the revision process.  I hope this feedback has been helpful.

Author Response

We are grateful for your consideration of this manuscript, and we also very much appreciate your suggestions, which have been very helpful in improving the manuscript. The comments are addressed below.

General comments:

Throughout this manuscript, there are opportunities to make the message clearer by improving the word choice, avoiding repetition, and correcting grammatical errors.

 In addition, I would suggest that overuse of abbreviations interferes with readability, and therefore, understanding.

DONE- We rewrote the article and we think it´s more clear now.

Was there any effort to blind the investigators to the condition during post-exercise blood pressure measurement? Given the hypothesis presented in the introduction and the confirmation of the hypothesis stated in the discussion, it would seem prudent to indicate some management for bias in the data.

DONE- Yes it was. We provide now that information in the blood pressure measurement procedures.

The results section needs work. First, the same data are presented in tables and figures.  This is redundant.  The text includes lengthy lists of individual changes, effect sizes, and significance.  Consider presenting these findings in a table with the data.  If the figures are to be used, please consider how you indicate significance.  For example, you may include one marker to indicate a significant difference to other conditions and one marker to indicate a difference from baseline (if this is important).  Four markers of significance for individual data points suggests inappropriate statistical analysis (too many individual comparisons).

DONE- We rewrote the results; we add one table with all the data.

The discussion lists individual findings from other papers. It would be helpful if these findings were grouped or summarized and, importantly, put into context relative to the findings in this study.  This is done well in lines 306-308, but not done so well elsewhere in the manuscript.

There are a few instances of ideas that should be suggestions rather than statements. For example, stating that your results “can be explained by xyz”, suggests you have proof of the mechanism.  I suggest alternative phrasing, such as “xyz may contribute to these findings” or “one mechanism that may have influenced this result is”.

DONE- We rewrote some parts of the discussion to avoid statements not supported by data.

Specific comments:

Title: consistent use of capital letters

DONE

Line 25: spaces after commas and format units

DONE – Line 25

Line 25-26: five visits on non-consecutive days – current wording is confusing

DONE – Line 26

Lines 40-44: be concise and specific, plus needs grammatical editing

DONE – Line 41-45

Line 46: phrasing

DONE

Line 49: prevention of normotensive individuals?

DONE – We rewrote.

Line 54: please avoid starting sentences with abbreviations

DONE

Line 54-55: phrasing

DONE

Line 61-62: phrasing

DONE

Line 64: do you mean they did not present their data or their data showed the intervention had no effect?

DONE- We rewrote Line 61-65

Line 68: normotensive individuals (plural)

DONE – Line 70

Line 72: double product?

DONE- We rewrote – Line 74

Line 85: inconsistent use of abbreviations – I’d encourage you not to abbreviate resistance exercise throughout the manuscript

DONE

Line 89: experimental design

DONE- We rewrote – Line 91

Line 95-96: volunteers were randomly assigned

DONE – Line 97

Lines 100-101: Sixteen normotensive men (tidy up units) with at least three years experience…

DONE- We rewrote – Line 103

Lines 103-111: this section needs to be reviewed

DONE- We rewrote – Line 105-114

Figure 1: were the warm up sets carried out for all exercises?

DONE

Line 168: check your equation

DONE: We Checked [systolic blood pressure + (2 X diastolic blood pressure)] / 3

Line 173-174:

DONE

Line 261-264: phrasing could be improved for clarity

DONE- We rewrote – Line 250-253

Line 265-266: this is repetitive

DONE

Line 283-285: rephrase for clarity

DONE- We rewrote

Line 286-292: this explanation needs to be tidied up

DONE- We rewrote

Line 306-308: This is well done.  This is a nice example of putting other research findings into context to help explain your own findings.  Please do more of this throughout your manuscript.

DONE

Line 341: interesting does not fit here

DONE

Round 2

Reviewer 2 Report

In my opinion, the changes made to the manuscript are not sufficient to make it suitable for publication.

In the first version, some results and statistical analyses were dubious: For example, the authors reported that in one of the experimental groups mean blood pressure significantly changed without any change in systolic and diastolic blood pressure. It was very weird since mean blood pressure was calculated by the equation: [systolic blood pressure + (2 X diastolic blood pressure)] / 3. I pointed this out in my first review. In the second version of the manuscript, the authors ignored this problem. I got asked to review the revised version and pointed this out again. Now, in the third version of the manuscript, the authors reported that data were miscalculated.  The English language is still a major issue.  I hope my comments on the first version of this manuscript can help to improve the manuscript. 

Author Response

We are grateful for your consideration of this manuscript, and we also very much appreciate your suggestions, which have been very helpful in improving the manuscript. We also thank the reviewers for their careful reading of our text. All the comments we received on this study of all reviewers have been attended into account in improving the quality of the article. 

Reviewer 3 Report

Thank you for your attention to my previous comments.  This version of the manuscript is an improvement upon the last.

There are still units that require formatting, for example the units for BMI are m2·kg-1 (the 2 and -1 are in superscript and there is a dot between the m2 and the kg-1 ).

Table 1 presents the data more clearly that the previous passages of text.  This is good.

There are still four markers of significance on individual data points in your graphs (Figure 2).  This is confusing and, I repeat, it introduces concern about the statistical analysis.

There is still work to be done, particularly in the discussion, summarizing the literature to put your findings into context.

Author Response

We are grateful for your consideration of this manuscript, and we also very much appreciate your suggestions, which have been very helpful in improving the manuscript. 

We send the corrected manuscript. It was corrected according to the editors suggestion and following the requests by reviewer 3, namely: i) correction of units (lines 25 and 102; ii) removal of significance markers (figure 2). We removed two symbols and we are now left with two significance markers that depict differences between treatments.

This manuscript is a resubmission of an earlier submission. The following is a list of the peer review reports and author responses from that submission.

Round 1

Reviewer 1 Report

The authors of the present study were tried to understand the effect of IPC on blood pressure following RE and based on the results the authors concluded that IPC helps in reducing BP following RE. 

 Here are my suggestions

1) The abstract is difficult to follow and its presentation must be improved. 

2) These studies were performed on young healthy adults and how does it have clinical relevance?. It is very difficult to correlate the present study results with patients with systemic arterial hypertension. 

Reviewer 2 Report

Dear authors,

I give you my congratulations for your interesting research, the methods and the presentation of the information during the entire paper are very good. However, I think that authors should perform the following suggestions:

I suggest changing the objective: “Thus, the objective was to verify the acute effect of IPC in a session of resistance exercises on systolic blood pressure (SBP), diastolic blood pressure (DBP) and mean blood pressure (MBP) of trained normotensive young individuals.” To “Thus, the objective was to verify the acute effect of IPC BEFORE a session of resistance exercises on systolic blood pressure (SBP), diastolic blood pressure (DBP) and mean blood pressure (MBP) of trained normotensive young individuals.  

Both introduction and discussion may be improved with some more references from the literature such as the following:

Acute effect of resistance exercises performed by the upper and lower limbs with blood flow restriction on hemodynamic responses. Journal of Exercise Physiology Online, 2016 Effects of pre-exhausting the biceps brachii muscle on the performance of the front lat pull-down exercise using different handgrip positions, 2014

Line 57 you refer that IPC improves performance but you don´t specified, I suggest to add some references to strength this sentence

Do inclusion criteria include performance variables?

Line 124 – can you refer a study that used your protocol of IPC.?

Line 143 and 144. Change to: Training load prescription was based in the 1RM evaluation [1]. The evaluations were performed in the 1st and 2nd visit to the laboratory.

I suggest changing the title replacing “exercises” by “exercise”.

In results I think that it´s necessary to include a figure with all the results to improve visibility of your research

Line 333 please add a comment to the health benefits of IPC on PEH

Reviewer 3 Report

This study was designed to examine the effects of acute ischemic preconditioning on post-exercise hypotension after a resistance exercise session. Healthy trained subjects performed three different protocols: Ischemic preconditioning followed by resistance exercise session (IPC + RE); SHAM ischemic preconditioning followed by resistance exercise session (SHAM + RE) and; resistance exercise only (RE). Arterial blood pressure was monitored before, immediately after and during 60 minutes after resistance exercise sessions (at 10 minutes intervals). The authors suggest that ischemic preconditioning potentiates the post-exercise hypotension caused by a resistance exercise session. In my opinion the manuscript is not well written and the results are not accurately interpreted.

Major comments:         

In 2.6 Statistical Analysis it is stated: “To test the normality of the data, the Shapiro-Wilk test, the Levene test for homosexuality…”. This type of mistake is unacceptable. It is even more serious considering that the manuscript was (presumably) read by 10 co-authors and this terrible mistake was not identified.

There are several grammatical/language mistakes in the text. The manuscript must be fully revised. Just a few examples (from many):

- Abstract, line 31: “3 series at 80% of 1 RM…” I guess the authors meant “3 sets”?

- Abstract, lines 34/35: “IPC+RE protocol showed significant reductions between SHAM+RE(p<0.05) and RE (p <0.05) in SBP, DBP and MBP”. What did the authors mean?

- Discussion, line 316: “Regarding the answers on DBP…”. Answers??? What did the authors mean?

The interpretation of the results is inaccurate. In the Abstract, lines 38/39, it is stated: “It can be concluded that the IPC before the RE session can potentiate PEH, although all protocols have shown PEH”. This is not what tables 1, 2 and 3 are showing. SHAM+RE did not promote any significant decrease in systolic blood pressure and only decreased diastolic blood pressure at 40 minutes post-exercise as compared to its respective baseline values. Moreover, RE reduced systolic blood pressure at 20, 30 and 40 minutes, did not affect diastolic blood pressure but reduced mean blood pressure at 50 and 60 minutes? Is that correct? It is weird since mean blood pressure was calculated by the equation: [systolic blood pressure + (2 X diastolic blood pressure)] / 3.

One sentence, many problems:

       Discussion, lines 334-336: “Moreover, IPC can attenuate the increase in peripheral vascular resistance after a session of RE caused by high intensity, thus generating PEH of great magnitude and duration also over the DBP”. First, was the peripheral vascular resistance measured in the present study? Second, it is not possible to understand the whole sentence. It must be rewritten.

Conclusion, lines 345-347: “Thus, the application of IPC can be considered a safe and efficient strategy for prescribing high-intensity RE with the purpose of preventing hypertension for different populations”. This kind of extrapolation must be avoided. Safety, efficiency, hypertension prevention and different populations: were these aspects investigated in the present study? Your conclusion must be based on data.

Minor comments:

RE is the control group of the study and should be presented first in the tables.

The abbreviation “BP” cannot be used for both blood pressure and bench press.